# Ferritin-Coated SPIONs as New Cancer Cell Targeted Magnetic Nanocarrier

**DOI:** 10.3390/molecules28031163

**Published:** 2023-01-24

**Authors:** Luisa Affatigato, Mariano Licciardi, Alessandra Bonamore, Annalisa Martorana, Alessio Incocciati, Alberto Boffi, Valeria Militello

**Affiliations:** 1Department of Physics and Chemistry-Emilio Segrè, University of Palermo, 90128 Palermo, Italy; 2Dipartimento di Scienze e Tecnologie Biologiche Chimiche e Farmaceutiche (STEBICEF), University of Palermo, 90123 Palermo, Italy; 3Department of Biochemistry-A. Rossi Fanelli, Sapienza University, 00185 Rome, Italy

**Keywords:** ferritin, SPIONs, coating, nanoparticles, cancer cell targeting

## Abstract

Superparamagnetic iron oxide nanoparticles (SPIONs) may act as an excellent theragnostic tool if properly coated and stabilized in a biological environment, even more, if they have targeting properties towards a specific cellular target. Humanized *Archaeoglobus fulgidus* Ferritin (HumAfFt) is an engineered ferritin characterized by the peculiar salt-triggered assembly-disassembly of the hyperthermophile *Archaeoglobus fulgidus* ferritin and is successfully endowed with the human H homopolymer recognition sequence by the transferrin receptor (TfR1 or CD71), overexpressed in many cancer cells in response to the increased demand of iron. For this reason, HumAfFt was successfully used in this study as a coating material for 10 nm SPIONs, in order to produce a new magnetic nanocarrier able to discriminate cancer cells from normal cells and maintain the potential theragnostic properties of SPIONs. HumAfFt-SPIONs were exhaustively characterized in terms of size, morphology, composition, and cytotoxicity. The preferential uptake capacity of cancer cells toward HumAfFt-SPIONs was demonstrated in vitro on human breast adenocarcinoma (MCF7) versus normal human dermal fibroblast (NHDF) cell lines.

## 1. Introduction

Nanotechnology is at the leading edge of the rapidly developing new therapeutic and diagnostic concepts in all areas of medicine. Among many drug delivery systems (DDSs), magnetic nanoparticles (MNPs) have gained important attention in the last decades [1]. MNPs are a class of theragnostic nanoparticles that can be manipulated under the influence of an external magnetic field. MNPs are commonly composed of magnetic elements, such as iron, nickel, cobalt, and their oxides [2]. They are classified by their response to an externally applied magnetic field [3]. The orientation of the magnetic moments in a particle allows identifying of different types of magnetism observed in nature. The magnetic properties of these particles are classified by the dependence of the magnetic induction B on the magnetic field H [4]. In most materials, the relation between B and H is linear: B = μ × H; where μ is the magnetic permeability of the particles. Iron oxide particles exhibit paramagnetism if μ > 1; and diamagnetism if μ < 1. One important advantage for the magnetic nanoparticles is their superparamagnetism which enables their stability and dispersion upon removal of the magnetic field as no residual magnetic force exists between the particles. Below approximately 15 nm, these particles are so small that the cooperative phenomenon of ferromagnetism is no longer observed and they magnetize strongly under an applied magnetic field but do not retain this property once the field is removed. Nanoparticles with this feature are called superparamagnetic particles and they are usually composed of a solid core made up of iron oxides (magnetite, Fe_3_O_4_, and/or maghemite, Fe_2_O_3_) coated with biocompatible polymers [5]. The versatility of the superparamagnetic iron oxide nanoparticles (SPIONs) allows the production of theranostic, multimodal and multifunctional devices that can be used for simultaneous drug delivery [6] and imaging [7,8], biomolecular tracking, and cellular labeling [9]. Although bare SPIONs exert some toxic effects, coated SPIONs have been found to be relatively nontoxic, they were approved by the US Food and Drug Administration (FDA) due to being quite benign toward humans [10]. To make SPIONs stable and suitable for biomedical applications, it is important to disperse the nanoparticles in water and modify their surface with small molecular surfactants or polymers. These surfactants or polymers protect the iron oxide core from agglomeration, to provide chemical handles for conjugation with biomolecules, and to reduce non-specific cell interactions. Additionally, studies have shown that the iron released from degrading SPIONs is metabolized by the body, reducing the potential for long-term cytotoxicity [11]. Various methodologies have been developed to synthesize SPIONs [12,13,14,15] and to functionalize them with specific coatings [16,17]. Targeting methods generally fall into one of two categories [18]: passive targeting, which relies on the physiological differences between cancerous and normal tissues; and active targeting, which relies on ligands conjugated to the surface of the SPIONs to recognize specific surface markers on cancerous tissue.

The biocompatible coating of SPIONs is essential for most biomedical applications since this increases the stability of the iron oxide core, preventing aggregates formation and allowing functionalization of the surface of the nanoparticles with targeting ligand [19]. There are many natural and synthetic polymers that can be used, such as dextran [20,21], starch [22], alginate [23,24], poly(D,L-lactide-co-glycolide) [25], and poly(ethylene-glycol) (PEG) [26,27], but also monoclonal antibodies, folic acid, biotin, transferrin, lactoferrin, albumin, insulin, growth factors, etc [28]. Among these, we used a specific ferritin, the Humanized *Archaeoglobus fulgidus* Ferritin (HumAfFt), as a biomimetic coating for SPIONs. 

HumAfFt is an engineered ferritin characterized by the peculiar salt-triggered assembly-disassembly of the hyperthermophile *Archaeoglobus fulgidus* ferritin and which is successfully endowed with the human H homopolymer recognition sequence by the transferrin receptor (TfR1 or CD71), overexpressed in many cancer cells in response to the increased demand of iron. Ferritin proteins have played an important role in recent years as smart nanocarriers for drug delivery due to their hollow cage-like structures and their unique 24-mer assembly [29,30]. Ferritin H-homopolymers have been extensively used as nanocarriers for different applications in the targeted delivery of drugs and imaging agents [31,32], due to their ability to bind the transferrin receptor (TfR1 or CD71), highly overexpressed in iron avid, fast replicating, tumor cells. On average it takes less than five minutes to complete an entire ferritin internalization cycle, which yields an approximate turnover rate of 20,000 ferritin molecules per cell per minute. This is a greatly advantageous feature of ferritin targeting because it allows a larger number of drug or other molecules to be internalized in the cells [33]. Virtually all proposed applications are based on the delivery of small therapeutic molecules or metal labels encapsulated within the human H-ferritin homopolymer with a procedure that entails subunit dissociation of the ferritin 24-mer at extreme pH values (<2.0 or >10.0) followed by neutralization and then the encapsulation of the small molecules [34,35]. This technique, however, is not amenable to the encapsulation of molecules that are highly sensitive to drastic pH changes. In contrast, the ferritin from *Archaeoglobus fulgidus* (AfFt) has emerged as an alternative to human ferritin homopolymers, since it requires mild cargo material encapsulation conditions in view of the unique self-assembly properties that entail divalent cation-driven assembly at neutral pH values. AfFt assembles in a distinctive tetrahedral geometry as a result of a particular packing between four hexameric units into a unique 24-mer structure, which results in the formation of four wide triangular pores (45 Å) on the protein shell. As such, AfFt has been proven to be the ideal scaffold to host molecules or nanoparticles within the internal cavity in a reversible manner [36,37,38].

*Archaeoglobus fulgidus* ferritin was genetically modified by grafting a 12 aminoacidic loop (BC loop in ferritin topology), typical of H-ferritin homopolymer, into the archaeal ferritin itself. The resultant chimeric protein, referred to as “humanized *Archaeoglobus fulgidus* ferritin” was shown to be able to interact with the extracellular moiety epitopes of the CD71 receptor of target cells in a similar way as the human H ferritin [39]. The engineered HumAfFt thus combines the versatility in assembly and cargo incorporation properties of AfFt with binding capabilities and cellular uptake properties of human H homopolymer.

The aim of this study was to use HumAfFt as a coating material for 10 nm SPIONs, in order to create a new magnetic nanocarrier able to discriminate cancer cells from normal cells and to be influenced by a magnetic field as well. The obtained complex (HumAfFt-SPIONs) was characterized in terms of HumAfFt and SPIONs content, morphology, size, and stability. Moreover, the preferential target of the HumAfFt-SPIONs towards cancer cells was demonstrated in vitro by biological assays.

## 2. Results and Discussion

### 2.1. Preparation and Characterization of HumAfFt-SPIONs

The effect of mutations on the MgCl_2_-mediated self-assembly of HumAfFt has been previously studied by size exclusion chromatography (SEC), in order to separate different possible oligomers according to their molecular size [40]. The increasing of the MgCl_2_ concentration allows the self-assembly of the dimers until they reach a stable polymeric structure around 500 kDa, corresponding to the expected 24-mer cage-like structure, at 50 mM MgCl_2_. The data highlighted that the chimeric HumAfFt maintained the cation-induced association/dissociation properties of archaeal ferritin and is possibly assembled into a 24-mer typical structure. The disassembled HumAfFt was incubated in the presence of the SPIONs (HumAfFt-SPION ratio = 1) and by restoring the concentration of MgCl_2_ to 50 mM (Figure 1).

The superparamagnetic behavior of HumAfFt-SPIONs at room temperature was observed macroscopically by attracting HumAfFt-SPIONs dispersed in water with an external magnet, as shown in Figure 2. Upon the magnet application outside the vial, HumAfFt-SPIONs were rapidly recovered and accumulated near the magnet, whereas a homogeneous nanoparticle dispersion was established again as the magnetic field was removed, suggesting superparamagnetic behavior and good physical stability of the coated nanoparticle. The above procedure resulted in a valid method for recovering and purifying HumAfFt-SPIONs, warranting the elimination of not assembled ferritin.

The thermogravimetric analysis of the solid residue of HumAfFt-SPIONs showed that the new nanosystem has a water content of 1% and a Fe_3_O_4_ content of 40%, measured as sample weight loss between 25 and 560 °C (Figure 3).

ATR spectra acquired for each sample are reported in absorbance mode in the range 4000–400 cm^−1^ (Figure 4) and present expected absorption peaks previously reported in the literature [41], confirming the composition of prepared HumAfFt-SPIONs. The peaks around 570 cm^−1^ and 630 cm^−1^ are attributed to the stretching vibrations from Fe-O and confirm the existence of nanoparticles with magnetite core [42]. Broad bands at the region 3000–3600 cm^−1^ corresponding to the vibrations of the hydroxyl group (O-H) are present in HumAfFt and HumAfFt-SPIONs spectra, in these compounds the water content is higher than in the SPIONs. Sharp peaks at 1650 cm^−1^ and at 1550 cm^−1^ in the HumAfFt spectrum are assigned to the N–H bending vibration of primary amines. The bands in the range 1450 cm^−1^ and 1100 cm^−1^ were attributed to C-H deformation vibrations and the C-O stretching, respectively. The appearance of these peaks in the HumAfFt-SPIONs spectrum suggests that the HumAfFt is coating the SPION surface.

The morphology and the size of prepared HumAfFt-SPIONs were evaluated by AFM analysis. AFM images (Figure 5) clearly showed a spherical particle population with a mean diameter size between 30 and 39 nm.

With the aim of confirming the presence of a ferritin coating around HumAfFt-SPIONs, an Energy Dispersive X-Ray (EDX) analysis was performed by SEM analysis [17] (Figure 6a–d). It revealed that HumAfFt-SPIONs aggregates are coated with oxygen and carbon-bearing organic material, such as ferritin; no iron is visible on the surface, demonstrating that the SPIONs constitute the core of the nanostructure. As a control, an elemental analysis of uncoated SPIONs was performed (Figure 6e). Differently, this analysis demonstrated that the same SPIONs without coating are composed of iron and oxygen in the ratio 89.8% and 10.2%, respectively.

#### Stability Studies of the HumAfFt-SPIONs

The stability of nanoparticles is an important parameter concerning applications of nanovectors. These should be stable in terms of size when dispersed in a physiological medium avoiding aggregation phenomena. Therefore, the stability of the HumAfFt-SPIONs was studied by measuring the hydrodynamic diameter and zeta potential by DLS analysis. As shown in Table 1, HumAfFt-SPIONs were found to be stable with low PDI maintaining a hydrodynamic diameter size below 150 nm in the hydrated state [43]. This characteristic was maintained for up to one month, when the analysis was repeated. Furthermore, the surface potential (Zp) of SPIONs was found to be negative (−31.3) before coating, then, decreases to −5.1 when they are coated by the HumAfFt. This finding is strictly coherent with the deposition of ferritin units around the SPIONs surface. The lowering of the polydispersity index (PDI) value further demonstrates the formation of a homogeneous and stable nanosystem.

### 2.2. In Vitro Biological Characterization of HumAfFt-SPIONs

#### 2.2.1. Cytotoxicity Assay

The evaluation of the cytotoxicity effect of a novel material yields important data for predicting the safety of the new system for in vivo applications. Toward this goal, the viability of two different cell lines was assessed in the presence of various concentrations of HumAfFt-SPIONs. Cell viability was estimated by the MTS assay using human breast adenocarcinoma (MCF7) and normal human dermal fibroblasts (NHDF) cell lines. MCF7 is a cancer cell line where the transferrin receptor is overexpressed, and it is used to investigate the anti-cancer activity of many drugs and the associated mechanism of action; NHDF is a non-tumoral cell line extensively used as a model of normal cells to screen cytotoxicity of novel compounds or carriers. These cells were incubated with uncoated SPIONs and HumAfFt-SPIONs at three different concentrations of 10, 50, and 150 μg/mL, for 4 and 24 h. The results, in terms of cell viability (%) as a function of sample concentration, are shown in Figure 7. These results show that the viability of both normal and cancer cells was always above 80% for all the tested concentrations, both after 4 and 24 h of incubation, indicating a good cytocompatibility of the new vector HumAfF-SPIONs. No statistical significance was revealed in the cell viability of all tested samples at the same concentration and incubation time (*p* > 0.05).

#### 2.2.2. Uptake Studies in Cell Culture Experiments by Fluorescence Microscopy

To further investigate the preferential targeting of HumAfFt-SPIONs into tumor cells, uptake experiments of cancer and normal cells were performed. MCF7 and NHDF cell lines were incubated with fluorescein labeled HumAfFt-SPIONs for 4 and 24 h; then particle internalization was investigated by fluorescence microscopy. Analyzing the obtained images (Figure 8), the internalization of HumAfFt-SPIONs is markedly higher in MCF7 cells in comparison with NHDF, endorsing the excellent capacity of HumAfFt-SPIONs to discriminate between cancer and normal cells. It was noticed that the ability of magnetic nanoparticles to discriminate their uptake between tumoral and non-tumoral cells strongly depends on the presence of HumAfFt that improves the nanoparticles internalization; this result is supported by the fact that the transferrin receptors are overexpressed on the cancer cell membranes [39]. Finally, the merge images (c, f, i, and l) confirmed that the HumAfFt-SPIONs predominantly have a cytoplasmic localization in MCF7 cells (images i and l) starting from 4 h post incubation.

## 3. Materials and Methods

### 3.1. Expression and Purification of HumAfFt

The gene encoding for a mutated ferritin from *Archaeoglobus fulgidus* was synthesized by Gene Art (ThermoFisher) and subcloned into a pET22b vector (Novagen) between the restriction sites NdeI and HindIII at 5′ and 3′, respectively. The recombinant plasmid was transformed into *Escherichia coli* BL21 for protein expression upon induction with 1 mM IPTG (isopropyl-β-D-1-thiogalactopyranoside) at OD_600_ = 0.6 for 16 h. Cells were harvested by centrifugation 3 h post-induction at 37 °C. Bacterial paste from 1 L culture was resuspended in 20 mM HEPES buffer, pH 7.5, containing 200 mM NaCl, 1 mM TCEP (tris(2-carboxiethyl)phosphine), and a complete TM Mini Protease Inhibitor Cocktail Tablet (Roche). Cells were disrupted by sonication and the soluble fraction was purified by heat treatment at 78 °C for 10 min. Denatured proteins were removed by centrifugation at 15,000 rpm at 4 °C for 1 h. The soluble protein was further purified by ammonium sulfate precipitation. The precipitated fraction at 70% ammonium sulfate was resuspended in 20 mM HEPES, 50 mM MgCl_2_, pH 7.5 and dialyzed versus the same buffer. As the final purification step, the protein was loaded onto a HiLoad 26/600 Superdex 200 pg column previously equilibrated in the same buffer using an ÄKTA-Prime system (GE Healthcare). Purified protein was concentrated to obtain the final protein preparation of 1 mg/mL and protein concentration was calculated by measuring the UV spectrum using an extinction coefficient of 32,430 M^−1^cm^−1^. Protein yield was ~40 mg/L culture.

### 3.2. Preparation of HumAfFt-coated SPIONs

Superparamagnetic iron oxide nanoparticles (SPIONs) (10 nm average particle size) water dispersion were purchased from Sigma Aldrich (Milan, Italy). The HumAfFt was disassembled by dialysis with demineralized water (Milli-Q quality) using a molecular porous membrane tubing MWCO: 3.5 kD (Spectral/Por Dialysis Membrane Standard RC Tubing) pH 7.4 in order to remove the MgCl_2_. Finally, the HumAfFt was reassembled in the presence of the SPIONs (HumAfFt-SPION ratio=1) and by restoring the concentration of MgCl_2_ to 50 mM. HumAfFt-SPION was collected and purified using an external magnet.

### 3.3. Characterization of the HumAfFt-SPIONs

Size and zeta potential values of HumAfFt, SPIONs, and the new complex HumAfFt-SPIONs were recorded by Dynamic Light Scattering (DLS) analysis (Malvern Zetasizer NanoZS, Worcestershire, UK). A complex concentration of 0.06 mg/mL at pH 7.4 was used for DLS and Z-potential measurements (mV), at 25 °C using an instrument fitted with a 532 nm laser at a fixed scattering angle of 173°. Thermogravimetric analysis (TGA) was performed using a LABSYS Evo STA TGA-DSC (Caluire, France) at heating rates of 7 °C/min between 30 °C and 500 °C and alumina crucibles were used in all experiments. Nitrogen purge gas was used with a flow rate of 5 mL/min.

An FTIR spectrometer (Bruker ATR FTIR, model ALPHA, Ettlingen, Germany) in attenuated total reflection (ATR) mode, equipped with a diamond measurement interface and controlled by OPUS software, was used to collect IR spectra. Spectra have been acquired in the range 4000–400 cm^−1^ with a resolution of 2 cm^−1^. Each measurement is the result of the average of 64 scans. The ATR diamond crystal was cleaned with 70% ethanol/water and a background measurement was performed between each sample. Sample was compressed against the diamond to ensure good contact between sample and ATR crystal.

Scanning electron microscopy (SEM) and Energy Dispersive X-ray (EDX) analysis were performed using a scanning electron microscope, ESEM Philips XL30 (Massachusetts, USA). Samples were dusted on a double-sided adhesive tape previously applied on a stainless steel stub. The HumAfFt-SPIONs were then sputter-coated with gold prior to microscopy examination.

Atomic force microscopy (AFM) analyses were performed in Tapping mode in air by a Bruker Dimension FastScan microscope (Santa Barbara, CA, USA) equipped with closed-loop scanners. Triangular FastScan A probes (resonance frequency = 1400 KHz, Tip radius = 5 nm) were used for acquisitions. The nanosystem was dropped onto a freshly cleaved mica surface as a thin layer aqueous dispersion (0.001 µg/mL^−1^) and dried overnight before observation.

### 3.4. In Vitro Biological Evaluations

#### 3.4.1. Cytotoxicity Assay

The cytotoxicity assays were carried out by the tetrazolium salt (MTS) assay, using a commercially available kit (Cell Titer 96 Aqueous One Solution Cell Proliferation assay, Promega). Human breast adenocarcinoma (MCF7) and normal human dermal fibroblast (NHDF) cell lines were used for the experiments.

MCF7 and NHDF cell lines were obtained from the Laboratory of Cell Cultures of Advanced Technologies Network Center (ATEN Center) of the University of Palermo. Cells were seeded at a density of 2.5 × 104 cells/well in 96-well plates in Dulbecco’s modified Eagle’s medium (DMEM, Euroclone, Italy) containing 10 vol% fetal bovine serum (FBS), 1 mM glutamine, 1% penicillin and 2% amphotericine B (0.25 mg/mL) (Sigma-Aldrich, Italy), under standardized conditions (95% relative humidity, 5% CO_2_ and 37 °C) and cultured for 24 h. Starting dispersions of SPIONs or HumAfFt-SPIONs (10, 50, 150 μg/mL) were prepared in the same medium, and 150 μL of each dispersion was added per well. Untreated cells were used as negative control. After 4 and 24 h of incubation, the medium was removed and cells were washed with DPBS. Then 150 μL of fresh medium and 20 μL of a MTS solution were added to each well. Plates were incubated for an additional 2 h at 37 °C. Then, the absorbance at 492 nm was measured using a microplate reader (PlateReader AF2200, Eppendorf, Hamburg, Germany). MTS assay was performed in triplicate and the viability was expressed as percentage obtained from the ratio between each sample with respect to their negative control (100% of cell viability).

#### 3.4.2. Uptake Studies by Fluorescence Microscopy

The uptake of HumAfFt-SPIONs was evaluated on culture of normal and cancer cell lines by fluorescence microscopy. NHDF and MCF7 cells were seeded at a density of 105 cell type/well into 8-well plates and cultured for 24 h in Dulbecco’s modified Eagle’s medium (DMEM, Euroclone, Italy) containing 10 vol% fetal bovine serum (FBS), 1 mM glutamine, 1% penicillin and 2% amphotericine B (0.25 mg/mL) (Sigma-Aldrich, Milan, Italy), under standardized conditions (95% relative humidity, 5% CO_2_ and 37 °C). The complex HumAfFt-SPIONs were labeled with the fluorescein sodium salt (Sigma-Aldrich) according to the manufacturer’s standard protocol. After 24 h, cells were incubated with 350 μL per well of cell culture medium containing HumAfFt-SPIONs at a final complex concentration per well of 150 μg/mL for 4 h and 24 h. Following, the cells were washed with DPBS and analyzed by fluorescence microscopy. The images were recorded using an Axio CamMRm (Zeiss, Jena, Germany). Untreated cells were used as negative control to set the auto-fluorescence.

#### 3.4.3. Statistical Analysis

The student’s two-tailed *t*-test was used to carry out statistical analysis. The criterion *p* < 0.05 was chosen to assign statistical significance. Data are the average of three experiments ± standard deviation.

## 4. Conclusions

In this work, a new magnetic nanocarrier targeted toward cancer cells was developed. In detail, an engineered ferritin was synthetized and used as a coating material for 10 nm SPIONs. For this purpose, the engineered ferritin was the humanized *Archaeoglobus fulgidus* ferritin (HumAfFt) characterized by the peculiar salt-triggered assembly-disassembly of the hyperthermophile *Archaeoglobus fulgidus* ferritin and which is successfully endowed with the human H homopolymer recognition sequence by the transferrin receptor (TfR1 or CD71), overexpressed in many cancer cells in response to the increased demand of iron. The newly engineered ferritin assembled in a distinctive tetrahedral geometry as a result of a particular packing between four hexameric units into a unique 24-mer structure, representing an ideal scaffold to host molecules or nanoparticles within the internal cavity. Thanks to the salt-triggered assembly mechanism and to the 24-mer typical structure of the ferritin, 10 nm diameter SPIONs were successfully coated with the HumAfFt. The new complex HumAfFt-SPIONs formation was confirmed by ATR-FTIR and EDX techniques. HumAfFt-SPIONs were found to be stable with low PDI and a hydrodynamic diameter size below 150 nm in the hydrated state and about 40 nm in the dry state. The obtained complex maintained the superparamagnetic property of SPIONs and the presence of ferritin coating was confirmed using the EDX analysis.

Biological studies on MCF7 and NHDF cell lines have shown that HumAfFt-SPIONs do not induce toxicity in cells even at high concentrations. Uptake assay confirmed the magnetic nanocarrier’s ability to preferentially accumulate into MCF7 cancer cells versus NHDF (non-tumoral cells), in agreement with the fact that the TfR1 is overexpressed in a cancer cell. After only 4 h post incubation, the HumAfFt-SPIONs predominantly have a cytoplasmic localization in MCF7 cells. Therefore, HumAfFt-SPIONs represent an excellent theragnostic tool with high stability and biocompatibility. The obtained results stimulate further exploration of cancer-targeted therapies.

## Figures and Tables

**Figure 1 molecules-28-01163-f001:**
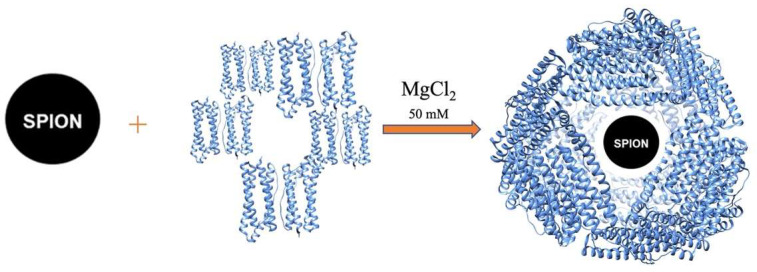
Schematic representation of the SPIONs coating with HumAfFt. The disassembled HumAfFt was incubated in the presence of the SPIONs and by restoring the concentration of MgCl_2_ to 50 mM.

**Figure 2 molecules-28-01163-f002:**
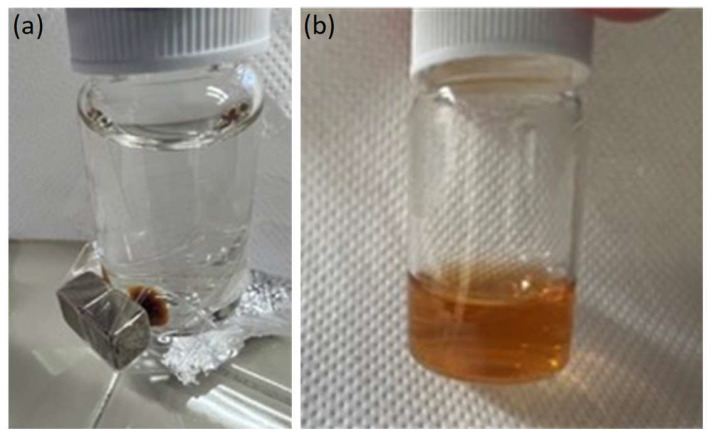
Pictures of a HumAfFt-SPIONs Milli-Q water dispersion during (**a**) and after (**b**) the application of an external magnet.

**Figure 3 molecules-28-01163-f003:**
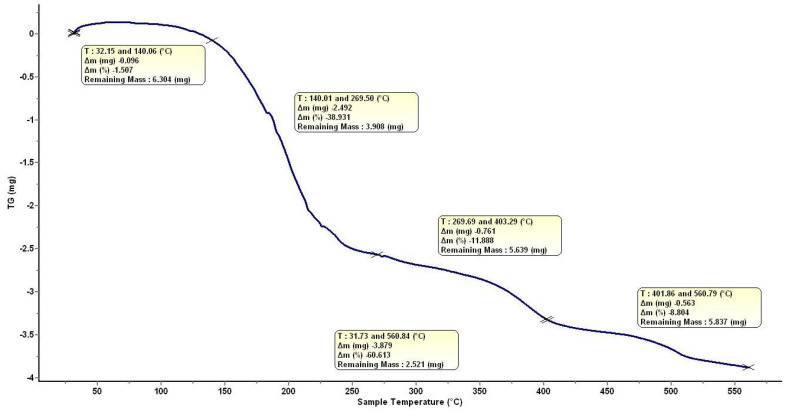
TGA analysis of HumAfFt-SPIONs.

**Figure 4 molecules-28-01163-f004:**
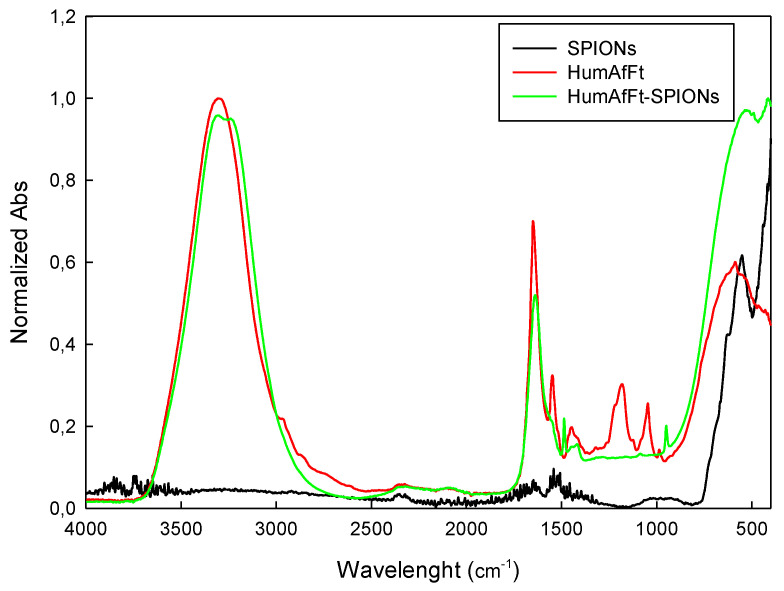
ATR-FTIR spectra of HumAfFt, SPIONs, and HumAfFt-SPIONs complex.

**Figure 5 molecules-28-01163-f005:**
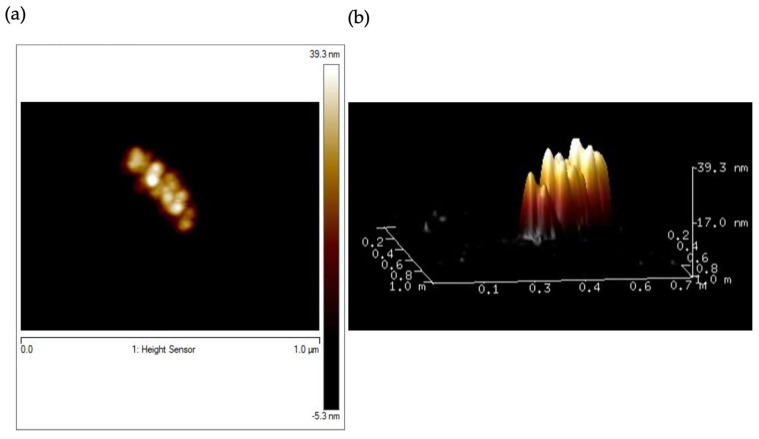
2D AFM micrograph (**a**) and 3D AFM image of HumAfFt-SPIONs (**b**).

**Figure 6 molecules-28-01163-f006:**
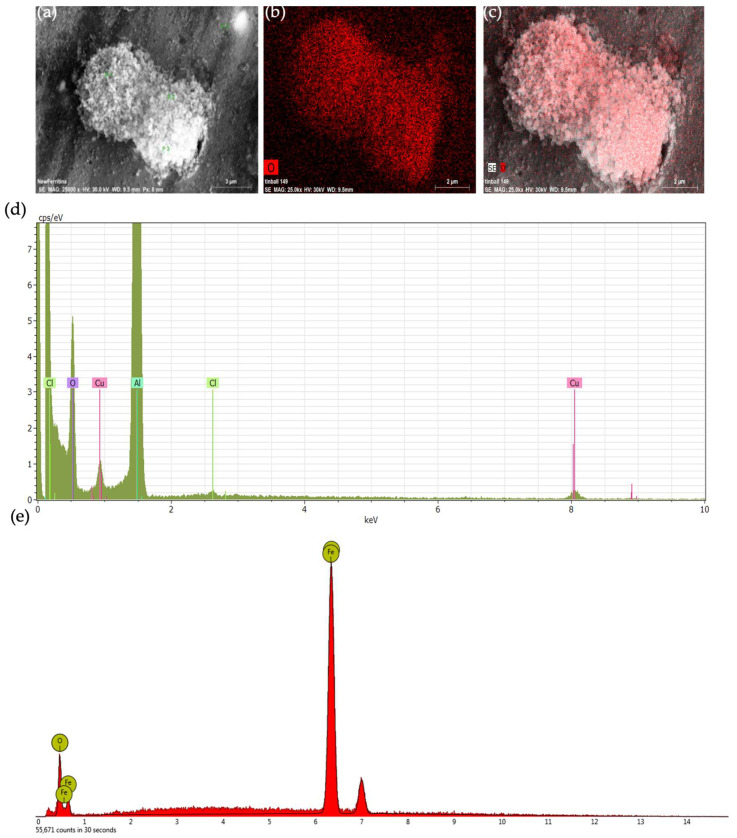
EDX analysis of the HumAfFt-SPIONs complex aggregates: (**a**) bright, (**b**,**c**) mapping of carbon and oxygen (red), (**d**) elemental analysis graph of HumAfFt-SPIONs, (**e**) elemental analysis graph of uncoated SPIONs.

**Figure 7 molecules-28-01163-f007:**
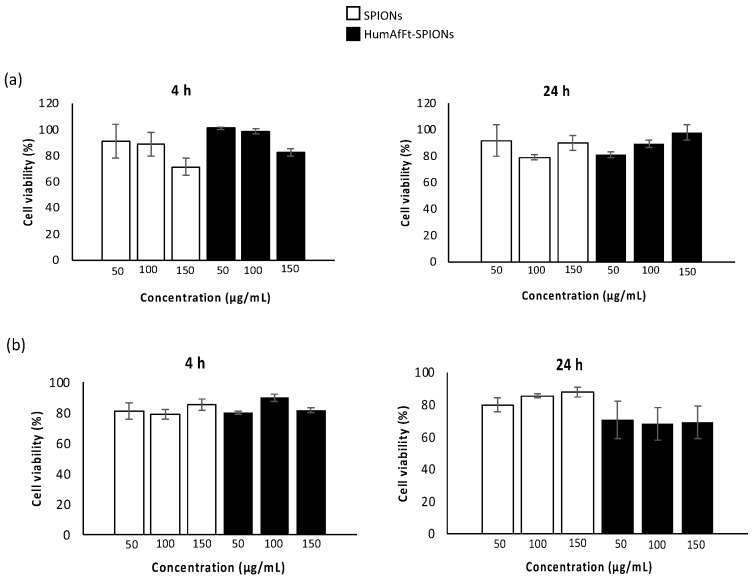
Cell viability of SPIONs (white) and HumAfFt-SPIONs (black) in MCF7 (**a**) and NHDF (**b**) cells after 4 and 24 h of incubation. Statistical significance (Student’s two-tailed *t*-test): *p* > 0.05 for all samples at the same concentration and incubation time.

**Figure 8 molecules-28-01163-f008:**
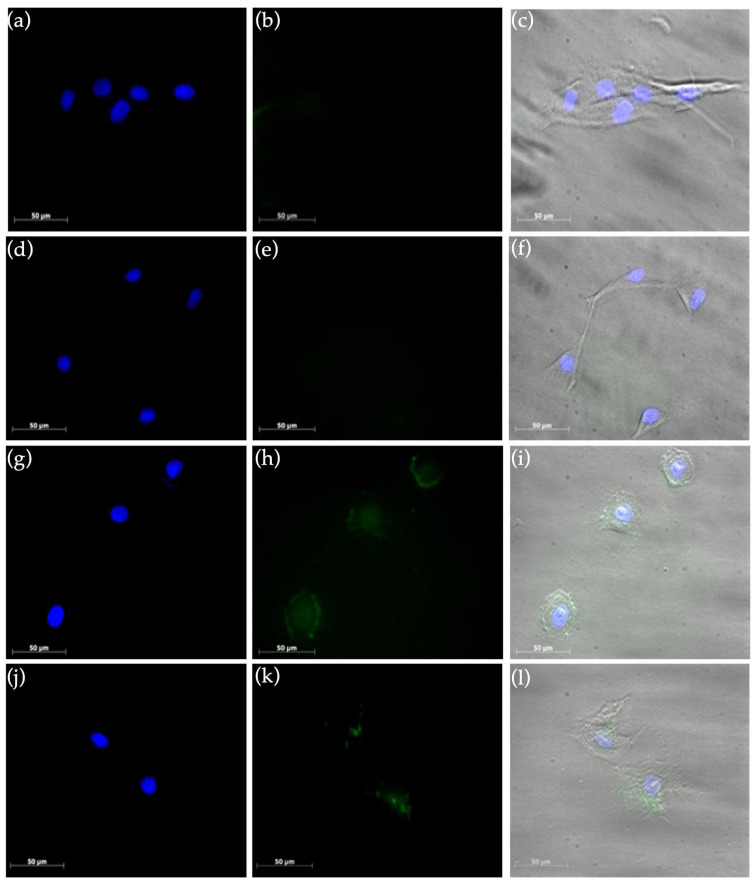
Uptake of HumAfFt-SPIONs in NHDF (images a–f) and MCF7 (images (**g**–**l**)) after 4 h (**a**–**c**,**g**–**i**) and 24 h (**d**–**f**, **j**–**l**) of incubation. Cell nuclei were stained with DAPI (blue in images (**a**,**d**,**g**,and **j**)) and Fluorescein-labelled HumAfFt-SPIONs are visualized in green (images (**b**,**e**,**h**,**k**)).

**Table 1 molecules-28-01163-t001:** DLS data (zeta average and polydispersity index) and Z-potential values of HumAfFt-SPIONs in Milli-Q water (0.06 mg/mL).

	Za (nm)	PDI	Zp (mV)
HumAfFt	20.42	0.392	−4.1
SPIONs	27.65	0.295	−31.3
HumAfFt-SPION	145.8	0.202	−5.1

## Data Availability

The data that support the findings of this study are available from the corresponding author upon reasonable request.

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
