# Peer review of "Ferritin-Coated SPIONs as New Cancer Cell Targeted Magnetic Nanocarrier"

_molecules, 2023, doi:10.3390/molecules28031163_

Round 1

Reviewer 1 Report

Report on Molecules-2149410

  In this manuscript the authors proposed the preparation of ferritin coated superparamagnetic iron oxide nanoparticles (SPIONs) which can act as new magnetic nanocarriers and could discriminate cancer cells from normal cells, and maintain the potential theragnostic properties of SPIONs. Further on, SPIONs were characterized in details in term of size, morphology, composition and cytotoxicity as well as in vitro investigation on human breast adenocarcinoma (MCF7) versus normal human dermal fibroblast (NHDF) cell lines. The characterizations were performed on DLS, FTIR, TGA, AFM images, EDX analysis, cytotoxicity assay and fluorescence microscopy.

The manuscript was written consistently and the work performed is of interest to the Molecules journal readership because it deals with very important, and highly investigated subject as development of magnetic nanoparticles for targeted biomedical, drug loading and drug delivery applications. After some minor revisions and negligible corrections in the text as described below, I recommend the manuscript to be accepted for publication.

 -1. What is actual biologic media? If the authors used for example PBS media, it should be mentioned in the text.

 -2. In Table 1 the sizes of nanoparticles were measured in water. Was there difference in their size in water and after incubation in the actual media.

 -3. How the authors can explain the big difference in the size of nanoparticles obtained by DLS measurements (145 nm) versus AFM analyses (39 nm)?

Author Response

Reviewer 1

 -1. What is actual biologic media? If the authors used for example PBS media, it should be mentioned in the text.

Authors thank the reviewer for this useful comment that help us to improve the manuscript.

The actual biological medium is DMEM, the same medium in which the cells were seeded. We clarified it in the experimental sections (3.4.1. Cytotoxicity assay and in 3.4.2. Uptake studies by fluorescence microscopy). Further details were added in the revised manuscript as well. The requested changes have been highlighted in red text in the revised version of the manuscript.

-2. In Table 1 the sizes of nanoparticles were measured in water. Was there difference in their size in water and after incubation in the actual media.

Actually, the sizes of nanoparticles were measured both in water and in the DMEM medium, and no significant size difference was evidenced. Size results recorded in DMEM medium were not reporrte, because, as expected, the quality report provided by the instrument (DLS) was not good, since DMEM composition influence scattering intensity.

-3. How the authors can explain the big difference in the size of nanoparticles obtained by DLS measurements (145 nm) versus AFM analyses (39 nm)?

We thank the Reviewer for this comment and give as the possibility to clarify this point.

The differences in the sizes of nanoparticles obtained by DLS versus AFM analyses depend on the measurement conditions. The DLS analysis gives information about the size of the hydrodynamic diameter of the hydrated nanoparticles, since the colloid interact with water molecules, generating a hydration layer; differently the AFM is carried out on dry sample and gives information about the size of the diameter without the hydrated layer. The above experience is already documented in literature (International Journal of Pharmaceutics 625 (2022) 122134. https://doi.org/10.1016/j.ijpharm.2022.122134).

Reviewer 2 Report

normal cell =is a non-tumoral cell line extensively used as model of normal cells to screen cytotoxicity of novel compounds or carriers.????? the normal skin cell line is not the same as the breast cancer line in terms of tissue source Why is it used?

What is the reason for choosing time in Cytotoxicity assay ?

Author Response

Reviewer 2

-1. Normal cell =is a non-tumoral cell line extensively used as model of normal cells to screen cytotoxicity of novel compounds or carriers.????? the normal skin cell line is not the same as the breast cancer line in terms of tissue source Why is it used?

The authors understand the reviewer's perplexity, however it is important to underline the fact that

the aim of this study is principally to demonstrate the ferritin-magnetic nanocarrier ability to preferentially accumulate into cancer cells than in non-tumoral cells. Actually we used two commonly used model cell lines both available in our laboratories: as reported in literature, MCF7 and NHDF are respectively a cancer cell line and a non-tumoral cell line used to investigate the cytotoxicity activity of new systems and drugs (Scialabba C, Puleio R, Peddis D, Varvaro G, Calandra P, Cassata G, et al. Folate targeted coated SPIONs as efficient tool for MRI. Nano Res. 2017 Sep 1;10(9):3212–27; Licciardi M, Scialabba C, Puleio R, Cassata G, Cicero L, Cavallaro G, et al. Smart copolymer coated SPIONs for colon cancer chemotherapy. Int J Pharm. 2019 Feb 10;556:57–67). Certainly, as suggested by the reviewer, the authors would have used the normal breast cell line if readily available. The authors intend to test the carrier systems on various tumor lines in a forthcoming study.

-2. What is the reason for choosing time in Cytotoxicity assay?

Authors chose the reported times in the cytotoxicity assay on the basis of the previous studies reported in literature that used the same cellular lines (Licciardi M, Scialabba C, Cavallaro G, Sangregorio C, Fantechi E, Giammona G. Cell uptake enhancement of folate targeted polymer coated magnetic nanoparticles. J Biomed Nanotechnol. 2013 Jun;9(6):949–64).